# Simultaneous pancreas and kidney transplantation: A qualitative study of partners' experiences

**Katie E. J. Hann**[1,2], **Marco Cinnirella**[1], **Clare Bradley**[1¤a], **Andrea Gibbons**[2¤b]*

**1** Department of Psychology, Royal Holloway University of London, Egham, United Kingdom, **2** Health Psychology Research Unit (HPRU), Health Psychology Research Ltd, Egham, United Kingdom

¤a Current address: Health Psychology Research Unit (HPRU), Health Psychology Research Ltd, Egham, United Kingdom
¤b Current address: Department of Psychology, University of Winchester, Winchester, United Kingdom
* andrea.gibbons@winchester.ac.uk

**Data Availability Statement:** De-identified interview transcripts have not been made available as they have been collected within a small group of participants who could be identifiable using basic

## Abstract

Chronic health conditions often affect the lives of family members as well as the patient themselves. The current study aimed to explore the experiences of partners of individuals with diabetes and chronic kidney disease (CKD) who received a simultaneous pancreas and kidney transplant (SPKT) to understand the wider impact of SPKTs. Eight partners of recipients of SPKT were interviewed about their experiences before and after the transplant. Interviews were transcribed verbatim and analysed thematically. Participants described how they navigated life with an unwell partner; sub-themes included a) living with pervasive worry, b) the challenge of enhanced responsibilities, and c) the buffering effect of social support. Diabetes complications, such as the experience of severe hypoglycaemia, particularly fuelled worry. Participants felt great relief after their partner's successful transplant but also faced certain realities around the potential for their partner's health to deteriorate again. The study highlights the impact of diabetes and CKD on patients' families and the wider benefit of transplantation, not just for the patient. The pancreas transplant, in addition to the kidney, relieved partners of their worry about hypoglycaemic events and the development of diabetes complications. Partners may benefit from being encouraged to seek support and to maintain their own health and wellbeing.

## Introduction

Diabetes is one of the leading causes of chronic kidney disease (CKD) [1, 2]. In the UK, those with type 1 diabetes who go on to develop kidney failure may be offered a simultaneous pancreas and kidney transplant (SPKT), which treats both conditions. Diabetes and CKD can have far-reaching negative impacts on patients' lives including quality of life and wellbeing [3–6] but transplantation is often beneficial in minimising the negative impact of the conditions [7–9]. It is important to consider the wider impact on significant others, as this may also affect their wellbeing, quality of life and, in turn, their ability to support the patient. Diabetes can

characteristics used in the analysis (e.g. age, gender etc.). The data also contain potentially sensitive patient medical information that may violate participant privacy. Participants did not consent to de-identified data being shared outside of the research team, so to do so would breach ethical approval. Ethical approval for this work was given by the East of England – Cambridge Central Research Ethics Committee (REC reference: 18/EE/0256), or the London - Queen Square Research Ethics Committee (REC reference: 18/LO/013). Requests for the data can be made to cambridgecentral.rec@hra.nhs.uk, quoting the reference REC reference: 18/EE/0256.

**Funding:** The author(s) disclosed receipt of the following financial support for the research, authorship, and/or publication of this article: This work was supported by a College Studentship funded by the Health Psychology Research Unit [code 010264-10 and P10370-10].

**Competing interests:** The authors have declared that no competing interests exist.

have a negative impact on other family members, especially if the individual with diabetes experiences severe hypoglycaemia, when blood glucose levels fall dangerously low [10, 11]. Partners have reported planning activities and mealtimes/diet to fit diabetes management, as they try to minimise the occurrence of diabetes complications and severe hypoglycaemia, all of which can result in anxiety and frustration [12–14].

Family caregivers of individuals with CKD (and particularly those receiving dialysis treatment) have also reported providing support which can impact aspects of their quality of life, such as their ability to socialise, work, plan, and travel as much as they would like. Family can also be affected by worry, fatigue, and uncertainty about the progression of CKD [15–19]. What little research has investigated the impact of kidney transplantation on family members has been quantitative, and none has focused specifically on SPKT recipients' significant others. Partners of kidney transplant recipients have reported less involvement in caregiving, fewer sexual relationship issues or social issues, and significantly better quality of life compared to partners of individuals on dialysis [20]. Partners/family caregivers of kidney transplant recipients have also reported better mental and physical health, well-being, and sleep, and less burden than those supporting individuals on dialysis and/or awaiting transplantation [21–24]. However, findings have not always been consistent. For example, Rodrigue et al. (2010) [25] found that partners of kidney transplant recipients reported better life satisfaction compared to partners of individuals awaiting kidney transplantation, but no other significant differences on other measures including the Short-Form 36 Health Survey [26], Profile of Moods States [27] and the Caregiver Strain Index [28]. The authors suggest this could be because partners continued to support the transplant recipients with their post-transplant treatment or comorbidities that continued to require care [25].

SPK transplantation has the potential to impact positively on the quality of life of patients by treating both diabetes and CKD. Whilst it was anticipated that those closest to SPKT recipients would also benefit from a successful transplant, this had not previously been studied. Qualitative methods can help to provide deeper insight into the impact of diabetes, CKD, and SPKT on significant others to provide an understanding in their own words of how and why family members are also affected. Therefore, the current study aimed to explore the experiences of SPKT recipients' partners to understand: (a) how partners are impacted by diabetes, CKD, and wait-listing for SPKT; (b) how partners support individuals with diabetes and CKD; (c) what impact SPK transplantation has on partners; and (d) how partners cope throughout the experience of wait-listing and SPK transplantation.

## Materials and methods

### Design

Qualitative semi-structured telephone interviews were used to explore the experiences of partners of SPKT recipients.

### Recruitment and procedure

Long-term partners (relationship of at least 12 months) were recruited from two studies assessing patient-reported outcomes in transplant recipients (including SPKT) attending UK transplant units [29]. Patients attending large urban transplant centres in the UK, who were at least six months post-SPKT, and were in a co-habiting relationship were invited to give the details of their partner and consent to them being contacted about an interview study. Of the 32 SPKT recipients from the studies, 22 reported having a partner and 17 gave consent for their partner to be contacted. Potential participants were then contacted directly by telephone to invite them to take part in an interview. A study invitation, information sheet, consent form,

and a brief questionnaire pack were posted to individuals who indicated willingness to consider taking part and to those who could not initially be contacted by telephone. Fifteen partners were invited to take part and eight returned signed consent forms and were subsequently interviewed between 1st July 2019 and 31st January 2021.

Participants were asked to complete a questionnaire which included questions about demographics including age, gender, education level, and employment. Interviews were conducted by telephone and followed a semi-structured interview topic guide (See S1 File). The research formed part of a PhD investigating quality of life and other patient reported outcomes of people receiving treatment for diabetes and CKD, and there was also a focus on the quality of life of the SPKT recipient in these interviews with their partners'. As all participants had been in their relationships with the SPKT recipients prior to transplantation, all were asked questions about their experiences before, during, and after the event. Broad open-ended questions were used to allow participants to tell their story, as well as more specific prompts when needed.

Interviews were conducted by KH, a PhD student with a Health Psychology MSc degree and previous experience of conducting telephone interviews. KH was not involved in the care of patients and not previously known to participants prior to taking part. The interviews lasted around 53 minutes on average, ranging from 39 to 75 minutes. With consent from the participants, the interviews were audio-recorded and then transcribed verbatim for analysis. A selection of the interview transcripts was reviewed by another member of the research team (AG) against the recordings for accuracy. The research plan was reviewed and approved by both the East of England–Cambridge Central Research Ethics Committee (REC reference: 18/EE/0256) and the London—Queen Square Research Ethics Committee (REC reference: 18/LO/0134). The standards for reporting qualitative research have been followed [30] (see S2 File).

## Analysis

The data were analysed using NVivo 11 software and Braun and Clarke's reflexive thematic analysis [31–33]. The researcher (KH) conducted and transcribed each interview before re-reading the transcripts whilst making notes on points of interest. Meaningful sections of the transcripts were coded and related codes were grouped together as preliminary themes. The analysis was inductive as it sought to develop themes that were grounded in the data. Through discussion within the research team, the preliminary themes were further revised, defined, and eventually named. The transcripts were re-read to ensure that the final themes reflected the experiences that participants described. A contextualist approach was taken, acknowledging that knowledge is context specific and is influenced by culture and other factors [34].

## Results and discussion

Table 1 presents the demographic characteristics of the eight participants, who included four men and four women aged 42 to 61 years. Participants were recruited from large urban transplant centres but the geographical location of participants' homes was not recorded. The time since the participants' partners received their SPKT ranged from 6–91 months. All but one participant reported that their partner had diabetes before they met them, but none had been diagnosed with kidney failure prior to the relationship. All participants' partners had been on insulin treatment for their diabetes prior to transplantation and three had received dialysis to treat their CKD. None of the participants considered themselves to be their partner's carer post-transplant.

The data were divided into two main themes (1) navigating life with an ill partner, and (2) relief and realities of life post-transplant. The first theme, navigating life with an ill partner, has three subthemes: a) living with pervasive worry, b) the challenge of enhanced responsibilities, and c) the buffering influence of support.

**Table 1. Characteristics of interview participants–Partners of SPKT recipients (*n* = 8).**

| Characteristic | |
|---|---|
| **Age*** *M (range)* | 52 (42–61) |
| **Sex: male** *n* (%) | 4 (50) |
| **Ethnicity** *n* (%) | |
| White British/European | 7 (88) |
| Asian British | 1 (12) |
| **Education** *n* (%) | |
| Basic qualification | 2 (25) |
| Higher qualifications | 5 (63) |
| Missing | 1 (12) |
| **Employment** *n* (%) | |
| Part-time | 4 (50) |
| Full-time | 4 (50) |
| **Length of relationship in years** *M (range)* | 24 (9–45) |
| **Months since partners SPKT** *M (range)* | 38 (6–91) |
| **Other chronic condition** *n* (%) | 4 (50) |

*Note*. *M*: mean *Missing data from one participant.

## Theme 1: Navigating life with an ill partner

This theme focuses on how participants' lives had changed since their partner became more seriously / chronically ill. There are three subthemes, living with pervasive worry, the challenge of enhanced responsibilities, and the buffering influence of support.

**Living with pervasive worry.** Participants reported feeling anxious and worried for their partner and their deteriorating health from diabetes, including the development of CKD and other long-term complications such as visual impairment. Participants were often particularly worried about their partner having severe episodes of hypoglycaemia or falling into a diabetic coma. The threat of worsening symptoms meant that some constantly monitored their partners health:

> *I think the biggest fear for me was for her losing her sight, I remember that thinking, mm, this is not good."* (Husband of SPKT recipient.)

> *"The first thing that I used to do in the morning was I used to actually lean over and make sure she was still breathing and that she hadn't had a diabetic coma in the night and died."* (Husband of SPKT recipient.)

In extreme situations, participants had found their partner unconscious or had been unable to wake them in the morning and needed to call for an ambulance. These severe episodes of hypoglycaemia left a lasting impression and fuelled worry:

> *"So she ended up having really quite serious, life-threatening crashes that er, ambulances and, you know, me injecting the hypo-stop, and doing that a couple of times. That was quite a scary situation."* (Husband of SPKT recipient.)

> *the stress of not knowing, not knowing whether she was going to be alive in the morning. It consumed me to the extent where it was all I thought about, erm, consciously and subconsciously.* (Husband of SPKT recipient.)

This worry extended not just to their partners' physical health, but also their ability to engage in activities. For example, participants discussed feeling anxious whenever their partner drove, was alone taking care of their children, or if participants had to be away from home for any length of time. One participant described how they always expected bad news: *whenever the phone rings it's like, oh no what's happened now*? (Husband of SPKT recipient.)

Not only were participants experiencing worry about complications from diabetes, but also the uncertainty and anxiety surrounding transplantation. Participants described having worried for their partner's life as they became increasingly unwell whilst on the waiting list or when they faced complications from the transplant operation in the initial post-transplant period:

*". . .just being in that place [the intensive care unit] is, you know, quite an intense realisation that this really is a matter of whether she lives or whether she dies. So that's quite, you get quite a strong realisation at that point."* (Husband of SPKT recipient.)

Participants managed their anxieties by engaging in concrete steps to minimise the risks to their partners. For example, participants whose partner had lost their awareness of hypoglycaemia became more vigilant and more adept at identifying when their partner's blood glucose levels were becoming low:

*"Whenever somebody's starting to go low they start, their behaviour becomes very repetitive, and also there was, something happened to her eyes. I can't explain it to you, but I could tell from her eyes that she was starting to go low."* (Husband of SPKT recipient.)

This monitoring extended to planning ahead to avoid hypoglycaemia altogether, such as ensuring they always carried something sugary whilst out. Some had also assisted their partner when they experienced hypoglycaemia by providing something to eat or drink or administering glucagon injections.

**Challenge of enhanced responsibilities.**   Not only were participants supporting their partner to take care of their health, such as attending appointments together and assisting them to manage their diabetes and avoid hypoglycaemia, they also described doing more of the driving, childcare, shopping, or chores around the home, often on top of continuing to work in a full-time or part-time job:

*"Taking the kids to school, doing all the, everything, every household chore, every household necessity was done by me as well as working at the same time. . ."* (Husband of SPKT recipient.)

The additional responsibilities and roles that several participants took on as their partner became more unwell impacted various aspect of their lives including their health, well-being, relationships, and work. For example, sleep patterns changed for one person because of their partners ill health:

*I think the biggest impact it had on me was, my sleep patterns changed. I became less able to sleep, I slept less deeply, erm, I was more tired, less effective at work. . . I suppose it's a, a kind of a, a natural unconscious response when, I suppose when you feel like you have a responsibility to protect somebody.* (Husband of SPK recipient)

For others, lack of time and the stress that they lived with whilst their partner was unwell meant that they had neglected aspects of their own health and wellbeing, such as taking regular

exercise. Some participants had stayed strong and well whilst their partner was very ill but later developed their own health problems:

> *"I've had a couple of health issues since, more recently. . . I think probably that, you know, you have to do the being the strong person and holding it all up and then. . . once the pressures off it all goes quietly to pieces a bit."* (Wife of SPKT recipient.)

A few felt that their life was "on hold" whilst their partner was on the transplant waiting list as they never knew when they would be called to the hospital for the transplant. Due to concerns over missing transplant opportunities, some had avoided traveling, especially abroad. Activities were also restricted as some worried about leaving their partner unassisted for long periods of time, limiting time that they focused on themselves. This meant that some had stopped doing things that they enjoyed, had avoided taking up new hobbies, or had missed social events to be around for their partner:

> *"I didn't have a social life, the things that [I] enjoyed doing I stopped doing."* (Husband of SPKT recipient.)

Another participant said:

> *"It isn't till you actually stop and think that, you know, how many things you've sort of turned down because it's just easier not to do it. I wouldn't get involved with going to classes with somebody and that sort of thing, you know, on a regular basis because it would always mean taking me away."* (Wife of SPKT recipient.)

For some, the nature of their relationship with their partner changed. Although none of the participants considered themselves to be a carer for their partner at the time of the interview post-transplant, they did indicate that they had assumed this role to some extent when their partner was unwell:

> *it's totally changed our relationship. It's been from in a relationship more to me being her carer, which impacts massively* (Husband of SPKT recipient.)

This resulted in some relationships becoming strained:

> *"she hadn't a good relationship with her diabetes. Very much in denial. And she was doing things that were not good for her health and me, she was very resistant to me to, to the changes that I was trying to implement. So, so that brought a lot of internal stress in the relationship. . ."* (Husband of SPKT recipient.)

The health conditions sometimes also negatively impacted sexual relationships, and some felt that their partner had become consumed by their health problems. This left some participants feeling forgotten and frustrated:

> *"everything was about him, so if I said, you know, I'm not feeling great today, he was always feeling worse than me. And it just, it got to a point where, you know, I had to say to him can it just be about me for today?"* (Wife of SPKT recipient)

For some, there was a sense of being in it together and the shared experience of their partner's health problems had brought them closer:

*"I know she was the patient but because we'd both experienced the whole process of it together, I think that that actually brought us closer together. And I think we just have a better understanding of each other because of that."* (Husband of SPKT recipient.)

Many participants indicated that their employers were supportive and allowed working from home and flexible working hours. However, balancing work and supporting the partner was challenging, especially when the partner's health declined. This meant that some participants missed opportunities to work, or worked odd hours to make up for missed time:

*"I think I just became less effective and the more busy I got in work, the more I had to do, the more I would work outside of normal hours. . . So I'd be working late and working weekends. . ."* (Husband of SPPK recipient).

Participants sometimes struggled because of their caring role on top of other responsibilities that they were juggling prior to and/or during their partner's initial recovery period posttransplant. This was particularly true for those also caring for children or elderly relatives as they often had little time to focus on their own wellbeing and described feeling burnt out:

*". . .I got quite spent, you know, at one point. I thought I'm just starting to feel a bit like an empty vessel, you know, I'm giving out and I've got no time to put anything back in."* (Wife of SPKT recipient.)

**The buffering influence of support.** The constant worry and increase in responsibility meant that support was often crucial for participants to cope with the situation. Participants often saw it as their responsibility to remain strong for the partner going through the health problems and wanted to protect them, so they did not always have space to express how they were feeling. This sometimes had a negative impact on participants' wellbeing and mental health:

*"I would say I supressed quite a lot through the actual process, because we were on autopilot. So you couldn't really let it come to the forefront, you couldn't deal with your emotions because you actually had to be aware of what was actually happening all around you. . ."* (Husband of SPKT recipient.)

Participants who were able to utilise other sources of support prior to transplantation and/ or during their partner's initial transplant recovery period reported this had helped them cope with the stress of their partners ill health. For some, this meant sharing responsibilities, such as having family and friends help with meal preparation or checking on the partner when they were away. This eased concerns about their partner's well-being whilst they were not around to help them and allowed some participants to continue with activities that were important to them:

*"While I was away we had to make sure that somebody rung her in the morning, made sure she was with us and all the rest of it. . ."* (Husband of SPKT recipient.)

Participants also benefited from receiving emotional support from friends and family which helped them to process and cope with what they were going through:

*"I've got a very good friend, I can talk to her and then, you know, we usually both get things off our chest and feel a lot better. . ."* (Wife of SPKT recipient.)

Support and/or information provided by other patients who were going through the same health problems and who had received a transplant was helpful. For example, some attended information events run by the renal teams prior to transplantation where they and their partner could speak to transplant recipients to find out about their experiences and what to expect. This gave them a point of reference and provided reassurance:

*". . .a few times I'd say to [my husband] oh, do you remember the lady that we spoke to and she said this, you know, oh yes of course. And it does help when you know what other people have experienced."* (Wife of SPKT recipient.)

Not all participants had sufficient sources of support available or did not feel confident that others could provide the necessary support, and this put more pressure on them:

*"So she'd become very reliant, even, to the support and that I was [providing], although there would've been other people that could've stepped in and done what I would've done. Because I lived with it every day, her family were nowhere near as switched on about it as I was, and I suppose that freaked her out."* (Husband of SPKT recipient.)

Even those who had support around them sometimes felt lonely and isolated during difficult periods such as bouts of declining health and the initial recovery period after their partner's transplant:

*"I think during the period of the transplant while she was in hospital, even though you've got family around you it does feel like a very lonely existence. . ."* (Husband of SPKT recipient.)

## Theme 2: Relief and the realities of life post-transplant

This theme focuses on the realities of life after SPK transplantation, which was mostly positive despite the often tough initial recovery period and some lingering concerns in the longer term.

Pressure continued for most during the initial post-transplant period, which was often a stressful time due to the frequency of appointments and risk of complications:

*". . . you got to come in twice a week and then that goes on a month, and then it goes to, like, once a week for the next month and then once a fortnight and on and on it goes. But that first month was hell."* (Husband of SPKT recipient.)

It was highlighted that more information on what to expect each day during the initial post-transplant recovery period would have been beneficial:

*"What would have been really useful in the hospital is if somebody gave him a list of, you know, on day one this is kind of how you're gonna feel and on day two, day seven, and just little insights that it's normal to feel like this and it's normal for this to be going on."* (Wife of SPKT recipient.)

However, after this worrying and stressful initial recovery period, participants felt great relief that their partner had received a successful SPKT. They were relieved to see their partner's health improve and some felt that their partner's life had been saved by the transplant:

*"That double transplant changed her life, gave her her life back. Gave her her energy back, gave her her health back, everything."* (Husband of SPKT recipient.)

Participants described returning to 'normal' life and having more freedom to do things, such as travel. They were glad that their partner had more dietary freedom, which made mealtimes easier for some. Similarly, some participants expressed relief that their partner no longer needed to follow a dialysis regimen, and they had regained space in their homes that had been taken up by supplies. Participants who had experienced relationship and/or sexual difficulties prior to their partner's transplant indicated that they had noticed improvements, and those with children found that their partner was able to take a more active parenting role and spend more time enjoying family activities. They were also happy to see their partner regain some of the independence that had been lost due to their poor health, such as being able to drive or go to work again:

*"He has now gone back to a part-time job, just over the last month or so, just a couple of shifts a week, but it's making him really happy. . ."* (Wife of SPKT recipient.)

Participants were relieved that they and their partner no longer needed to worry about diabetes management and the risk of further diabetes complications:

*"It was just amazing that something that had dominated his life for so long could be sorted out. . . it's just made such a difference. You know? That he doesn't have to be terrified of diabetic eye damage getting any worse or all the awful things."* (Wife of SPKT recipient.)

Participants no longer worried that their partner would experience severe hypoglycaemia and felt more able to take part in activities together:

*"Not having to worry about that [hypoglycaemia] now, is, yeh, that's a huge difference and I think it has been the biggest advantage to her having the transplant, the kidney and pancreas together, meaning she's not diabetic anymore, that has transformed her and transformed how we do things."* (Husband of SPKT recipient.)

The fear that something bad would happen to their partner when they were not around to help, such as severe hypoglycaemia, had also eased post-transplant. This relief and having more time for themselves allowed participants to refocus on their own health and well-being. Some had taken time to improve their mental health and/or fitness and were able to take part in activities that they enjoyed again:

*"Whenever you were going to do something you enjoyed, you could actually enjoy it because you weren't constantly worrying about what was happening at home, or what was happening to the wife. Whenever I was going to do things, I was able to do it for longer. So it just gave you that freedom that I hadn't had for years."* (Husband of SPKT recipient.)

Those who were interviewed around six months after their partner's transplant were still adjusting to life post-transplant. In particular, participants interviewed during the COVID-19 pandemic in 2020/2021 indicated that the restrictions of the pandemic prevented them from experiencing the full benefits of their partner's transplant. For example, they had not yet been able to travel abroad. Some felt the restrictions of the pandemic had given their partner time to rest and had raised awareness of good hygiene practices (which is especially important for

those on anti-rejection medications that supress the immune system). Participants had hope and looked forward to doing more in future, once the restrictions of the pandemic lifted:

*"And now we've got hope, we can go away, we can do normal things again. You know? Only 'cause this [Covd-19 pandemic] has been an exception, otherwise we would have been away on holiday enjoying ourselves."* (Husband of SPKT recipient.)

Although the worry of diabetes-related complications had lessened, some participants were still very aware that their partners were at risk of deteriorating health. Participants were careful not to pass on infectious illnesses to their partner and this was particularly relevant to those who were interviewed during the COVID-19 pandemic in 2020/2021. Whilst some had become used to living with restrictions prior to transplantation and during the recovery period, participants were aware that their partner was particularly vulnerable to the virus due to their suppressed immune systems:

*"It [COVID-19] was really, really scary at first. It was just, oh gosh, actually [my partner] is high risk, this is, you know, quite severe, he's in that top group and, you know, I need to be extra careful for his health. . ."* (Wife of SPKT recipient.)

Some were also concerned about how long their partner's graft would work and worried that it might suddenly stop working one day. This concern was common to both participants whose partner had only had the transplant for around six months and those whose partner had the transplant for years, but was not as pervasive as the worry they lived with pre-transplant:

*". . .because of all the pressure that has been on his body I do think, you know, could he wake up one day we're back to square one or his body rejects it or it's not working."* (Wife of SPKT recipient.)

## Discussion

This qualitative study explored the experiences of partners of SPKT recipients and found that participants often struggled to navigate life with a chronically ill partner prior to transplant. Participants experienced worry that influenced all aspects of their lives, from taking care of their ill partner, to concerns for the future. Participants' quality of life was impacted in numerous ways, from taking on more responsibilities for childcare and work, to changes in their relationships. The need for support for both participants and their partners was often reported to be crucial. Post-transplant, participants felt an immense amount of relief, and a greater sense of freedom both for themselves and their partner. At the same time, post-transplant was not always a fully positive experience, as several participants still held some concerns and worries for the future.

Taking the sub-theme of pervasive worry, some participants were concerned that their partner would experience life-threatening hypoglycaemia which was fuelled by frightening past experiences. Research into fear of hypoglycaemia has mainly focused on patients themselves or on parents of children with type 1 diabetes [35], however hypoglycaemia has been highlighted as a key concern in other qualitative research with partners of individuals with diabetes and can lead to watchful behaviours [10, 11]. A study which developed a measure of diabetes distress for partners found that around 30% reported diabetes-related distress and higher scores were associated with younger age, being female, number of previous severe hypoglycaemic

episodes, lower relationship satisfaction, less satisfaction with diabetes knowledge, and less comfort and satisfaction with their partner's healthcare providers [36]. As some evidence suggests that CKD is a risk factor for more frequent hypoglycaemia [37], partners of those awaiting an SPKT may particularly benefit from advice and support in this area.

Participants often experienced enhanced responsibilities as their partners health deteriorated and they sometimes prioritised their partner over their own health and well-being. This manifested in various ways, including limiting time spent on leisure and social activities. These findings are consistent with previous research [12, 38]. Furthermore, some participants had suppressed their feelings and kept concerns to themselves during difficult times as they tried to remain strong for their partner. This is sometimes referred to as protective buffering and is a type of relationship-focused coping [39] which extends Lazarus and Folkman's (1984) stress and coping theory and suggests that people will use certain strategies to try to maintain relationships [40]. Protective buffering is considered to be less adaptive as it can negatively impact the individual and their satisfaction with the relationship. On the other hand, active engagement, which entails being involved in discussions and problem solving about the health condition together, is considered a more adaptive method of relationship-focused coping [39, 41]. Worry and stress associated with having a partner with a chronic health problem may increase the risk of anxiety and depression [42]. Persistent stress may also put caregivers at risk of developing health conditions themselves, particularly if they neglect aspects of a healthy lifestyle, such as sufficient exercise [43, 44]. As maintaining independence and taking part in enjoyable leisure activities can be especially beneficial for individuals in caring roles [38], partners need to be actively encouraged to make time for their own well-being and health needs.

Participants reported support from others, and how that helped to minimise the impact of their partners' ill-health on such things as their own leisure activities and enabled them in some instances to continue to work. Social support is negatively associated with levels of burden [45] and depression reported by caregivers of individuals on dialysis [46, 47]. Healthcare professionals could explore ways in which patients and their partners could use their support network to ensure that partners are able to attend to their own health and well-being needs. Some partners may also benefit from being signposted to sources of psychological support as this is not available to them through the renal/transplant unit. Peer support from those living with the same health conditions or those who have received an SPKT can also be a useful source of information and reassurance for both patients and their partners [48, 49]. Individuals may want to access peer-support at various times throughout the progression of CKD and the process of receiving a transplant and need to be provided with opportunities to speak with other patients, especially as some may not request support themselves [49].

SPK transplantation resulted in great relief and more freedom for both participants and partners, highlighting the wider benefit of this type of transplant. Little research has considered the impact on partners, though one such study found that partners of individuals with a kidney transplant are less likely to feel burdened than they were post-transplant [24]. A particular benefit of the SPKT that is not seen in kidney only transplants, is that participants were no longer worried that their partner would experience severe hypoglycaemia and long-term diabetes complications. They were also able to enjoy more dietary freedom together with their partner. These findings are unique to the current study as research has not previously explored experiences of partners of SPKT recipients.

It should be noted that some participants in the current study were interviewed during the COVID-19 pandemic during which time the UK population lived with enforced social restrictions. Some participants interviewed at this time had not yet been able to experience the full benefits of their partner's transplant. Whilst participants were concerned about their partner's susceptibility to the virus, they had already adjusted to living with restrictions due to their

partner's illness. Consistent with findings from previous research with partners of kidney transplant recipients [20], some participants worried about how long their partner's transplant would last and whether they might become unwell again. However, this did not seem to be as pervasive a worry as that experienced prior to transplantation.

### Study limitations

The current sample was small, in part due to practical challenges of recruiting participants. Although larger samples are often considered more valid, it should be noted that the interviews provided relevant and rich accounts of participants' experiences and provided useful insight into the wider impact of chronic illness and transplantation. Concepts such as data saturation, are increasingly being acknowledged as problematic for studies such as this one that used reflexive thematic analysis, because it is inconsistent with the interpretive and subjective nature of the analysis [50]. Moreover, the findings complement those of previous research, which suggest that they are credible even with a smaller sample.

Having a smaller than average sample also means that it is possible that there may have been a selection bias in those who chose to take part. Previous research found that partners of kidney transplant recipients who reported greater relationship satisfaction were more likely to take part in research than the partners of those with poorer relationship satisfaction [51]. The current study did not include individuals in newer relationships or those who experienced a complete breakdown in the relationship, whose experiences may have been very different. Whilst the study aimed to explore partner's experiences as individuals, future research could include both members of each couple and investigate how couples cope together (dyadic coping) throughout the process of SPK transplantation to help identify ways in which couples could maintain relationship satisfaction. Furthermore, the current study did not investigate gender differences as this was beyond the scope of the research. Exploration of differences in experiences between men and women could help to identify whether there are differing support needs. Although participants were all recruited via large urban transplant centres across the UK, data related to where participants lived were not recorded. It is possible that those who live in rural areas further from the transplant centres may experience greater challenges to obtaining support. Further research could consider how rural vs. urban living may influence people's experiences, and whether or not support needs to be tailored to take this into account.

The authors are mindful that their experience as health psychology researchers and experience of conducting renal research may have influenced their interpretation of the data. Reflexive journals were kept, and authors AG and KH discussed the themes in depth.

### Conclusion

This is the first qualitative study to report on the experiences of partners of SPKT recipients. Pervasive worry was common, as was the need to take on enhanced responsibilities, resulting in sometimes neglecting their own health and well-being. Partners need to be encouraged to continue to pursue their own activities and to maintain their well-being and quality of life. They may also benefit from additional support in dealing with hypoglycaemia and other complications of diabetes, which lead to this heightened sense of worry. The initial few months after transplantation can be especially worrying and difficult, and partners may benefit from additional support and information on what to expect at this time. Patients and their partners need to be sign-posted to support offered at the renal/transplant unit or through relevant charities. However, SPK transplantation improved the quality of life of partners as well as SPKT recipients, as it relieved worry and enabled them to live more freely and refocus on their own health and well-being.

## Supporting information

**S1 File. Interview topic guide.**
(DOCX)

**S2 File. SRQR checklist.**
(DOCX)

## Acknowledgments

We thank the participants for giving their time to take part in this research. We would also like to thank Mr Gabriel Oniscu, Professor Christopher Watson, Dr Rachel Hilton and the nurses who helped to recruit patients to the original patient-reported outcomes studies which made the current research possible.

## Author Contributions

**Conceptualization:** Katie E. J. Hann, Marco Cinnirella, Clare Bradley, Andrea Gibbons.

**Data curation:** Katie E. J. Hann.

**Formal analysis:** Katie E. J. Hann, Marco Cinnirella, Andrea Gibbons.

**Funding acquisition:** Katie E. J. Hann, Clare Bradley.

**Investigation:** Katie E. J. Hann.

**Methodology:** Katie E. J. Hann, Clare Bradley, Andrea Gibbons.

**Project administration:** Katie E. J. Hann.

**Resources:** Marco Cinnirella, Clare Bradley, Andrea Gibbons.

**Supervision:** Marco Cinnirella, Clare Bradley, Andrea Gibbons.

**Writing – original draft:** Katie E. J. Hann.

**Writing – review & editing:** Katie E. J. Hann, Marco Cinnirella, Clare Bradley, Andrea Gibbons.

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
