## [Decision Letter · Decision Letter 0]

5 Jun 2024

PONE-D-23-40586Simultaneous pancreas and kidney transplantation: A qualitative study of partners’ experiencesPLOS ONE

Dear Dr. Gibbons,

Thank you for submitting your manuscript to PLOS ONE. After careful consideration, we feel that it has merit but does not fully meet PLOS ONE’s publication criteria as it currently stands. Therefore, we invite you to submit a revised version of the manuscript that addresses the points raised during the review process.

We look forward to receiving your revised manuscript.

Kind regards,

Vanessa Carels

Staff Editor

PLOS ONE

 [The author(s) disclosed receipt of the following financial support for the research, authorship, and/or publication of this article: This work was supported by a College Studentship funded by the Health Psychology Research Unit [code 010264-10 and P10370-10].].  

Please include this amended Role of Funder statement in your cover letter; we will change the online submission form on your behalf."

Reviewers' comments:

Reviewer's Responses to Questions

**Comments to the Author**

1. Is the manuscript technically sound, and do the data support the conclusions?

Reviewer #1: Yes

Reviewer #2: Yes

2. Has the statistical analysis been performed appropriately and rigorously? 

Reviewer #1: N/A

Reviewer #2: N/A

3. Have the authors made all data underlying the findings in their manuscript fully available?

Reviewer #1: No

Reviewer #2: Yes

4. Is the manuscript presented in an intelligible fashion and written in standard English?

Reviewer #1: Yes

Reviewer #2: Yes

5. Review Comments to the Author

Reviewer #1: Thank you for the chance to review this paper. It represents important perspectives of partners in relation to pancreas and kidney transplants. I have written a couple of questions and points of clarification below. Best wishes forward!

Introduction

The study is introduced with a relevant literature review and clear rationale for the study. The structure and logic of the text is helpful for readers.

Methods

The research process is clearly described and using the SRQR checklist is helpful. I have a question about recruitment, 15 partners were invited to participate, and 8 decided to do so. How did you decide that the 15 and the 8 were sufficient? Thinking of the four research questions, it seems it would have been a benefit to have a few more participants.

The interview guide is extensive and seems very structured. The concept quality of life is central in several questions. In the introduction, quality of life is mentioned once, as is well-being, health and life satisfaction. Perhaps other types of questions and foci would have generated different findings.

Findings

The findings are presented in six themes. The second theme is presented with two sub-themes, one covering two pages and one about ¼ of a page. I am uncertain about the reasoning behind this choice. In my reading it comes across as confusing and makes me wonder about balance in presentation of findings. Do the sub-themes contribute in a significant way?

The theme “Got on with it” is also quite short, perhaps it would make sense to merge it with another theme. My general impression of the findings is that they could be condensed and refined further for greater balance.

Discussion

The discussion dives straight into highlighting a few of the findings, fear of hypoglycemia, prioritizing their partner before themselves, getting on with it and relief after the transplantation. Please consider signposting this at the start of the discussion; it would be helpful for readers. Perhaps starting with a brief summary of the main findings could be useful.

It may also be helpful to consider the main message of the discussion; what should readers take or learn from these findings. Parts read a bit repetitive in comparison to the results.

The study limitations would benefit from considering a wider scope of the research process. At present only selection bias and gender are commented on. For example, trustworthiness and credibility of the findings could be elaborated.

Conclusion

The conclusion highlights health and well-being of the participants. Quality of life is not mentioned, which is a bit tricky as it is frequently used in the interview guide.

Reviewer #2: An important interview study with patners/caregivers of people requiring a SPK transplant. Some suggestions for improvement include:

1. This is a small sample size and I believe that needs to be acknowledged in the limitations paragraph.

2. Could you please clarify if all the partners where co-habitating with the person with kidney failure, as this may alter their experiences.

3. Was data saturation reached. I realize you only interviewed the 8/15 who responded but I want o know in the main body of the paper did you reach data saturation?

4. Findings - your themes and subthemes could do with another revision. They are very matter of fact and not conveying the true nature of the theme ie Living with worry- this could be improved to highlight the tre nature of the theme by including persistent or pervasive. Alos some of the themes have large bodies of writing that I feel highlight some subthemes that are important - impact on partners work for an example. Pleases revise and deduce whether further subthems could be highlighted.

5 For contextual assistance was this study conducted in city or rural areas as that does impact burden for partners and highlights further issues.

Thanks

6. PLOS authors have the option to publish the peer review history of their article (what does this mean?). If published, this will include your full peer review and any attached files.

Reviewer #1: **Yes: **Anna Klarare

Reviewer #2: No

---

## [Author Response · Author response to Decision Letter 0]

1 Oct 2024

Editorial team comments:

We have checked and amended the formatting of the manuscript so that it now complies with the style requirements of PLOS ONE. These amendments have not been highlighted in the tracked change document to allow for greater readability.

2. …Please state what role the funders took in the study. Please include this amended Role of Funder statement in your cover letter; we will change the online submission form on your behalf."

We have amended the Funding statement to include the information about the role of the funders.

3. We note that you have indicated that there are restrictions to data sharing for this study… If there are ethical or legal restrictions on sharing a de-identified data set, please explain them in detail (e.g., data contain potentially identifying or sensitive patient information, data are owned by a third-party organization, etc.) and who has imposed them (e.g., a Research Ethics Committee or Institutional Review Board, etc.). Please also provide contact information for a data access committee, ethics committee, or other institutional body to which data requests may be sent. We will update your Data Availability statement on your behalf to reflect the information you provide.

The de-identified interview transcripts have not been made available as they have been collected within a small group of participants who could be identifiable using basic characteristics used in the analysis (e.g. age, gender etc.). The data also contain potentially sensitive patient medical information that may violate participant privacy. Participants did not consent to data being shared outside of the research team, so to do so would breach ethical approval. The ethics committees concerned were the East of England – Cambridge Central Research Ethics Committee (REC reference: 18/EE/0256), or the London - Queen Square Research Ethics Committee (REC reference: 18/LO/0134) but we suggest that it would not be appropriate to provide wider access to these data. Requests for the data can be made to cambridgecentral.rec@hra.nhs.uk, quoting the reference REC reference: 18/EE/0256.

We have checked the reference list and have not identified any papers that have been retracted. We have included one additional reference to support points made in response to the reviewers’ comments on data saturation. This can be seen in the tracked change version.

---

## [Decision Letter · Decision Letter 1]

4 Nov 2024

Simultaneous pancreas and kidney transplantation: A qualitative study of partners’ experiences

PONE-D-23-40586R1

Dear Dr. Gibbons,

We’re pleased to inform you that your manuscript has been judged scientifically suitable for publication and will be formally accepted for publication once it meets all outstanding technical requirements.

Kind regards,

Edward Zimbudzi

Academic Editor

PLOS ONE

Additional Editor Comments (optional):

Reviewers' comments:

Reviewer's Responses to Questions

**Comments to the Author**

1. If the authors have adequately addressed your comments raised in a previous round of review and you feel that this manuscript is now acceptable for publication, you may indicate that here to bypass the “Comments to the Author” section, enter your conflict of interest statement in the “Confidential to Editor” section, and submit your "Accept" recommendation.

Reviewer #1: All comments have been addressed

2. Is the manuscript technically sound, and do the data support the conclusions?

Reviewer #1: Yes

3. Has the statistical analysis been performed appropriately and rigorously? 

Reviewer #1: N/A

4. Have the authors made all data underlying the findings in their manuscript fully available?

Reviewer #1: Yes

5. Is the manuscript presented in an intelligible fashion and written in standard English?

Reviewer #1: Yes

6. Review Comments to the Author

Reviewer #1: Thank you for your meticulous revisions and thoughtful responses. Your study is a valuable addition with rich information about partners' experiences related to transplantation. Best wishes forward!

7. PLOS authors have the option to publish the peer review history of their article (what does this mean?). If published, this will include your full peer review and any attached files.

Reviewer #1: **Yes: **Anna Klarare

---

## [Editor Report · Acceptance letter]

6 Nov 2024

PONE-D-23-40586R1 

PLOS ONE

Dear Dr. Gibbons, 

I'm pleased to inform you that your manuscript has been deemed suitable for publication in PLOS ONE. Congratulations! Your manuscript is now being handed over to our production team.

Kind regards, 

on behalf of

Dr. Edward Zimbudzi 

Academic Editor

PLOS ONE